# OFFLINE PRE-TRAINED MULTI-AGENT DECISION TRANSFORMER

## ABSTRACT

Offline reinforcement learning leverages static datasets to learn optimal policies with no necessity to access the environment. This is desirable for multi-agent systems due to the expensiveness of agents' online interactions and the demand for sample numbers. Yet, in multi-agent reinforcement learning (MARL), the paradigm of offline pre-training with online fine-tuning has never been reported, nor datasets or benchmarks for offline MARL research are available. In this paper, we intend to investigate whether offline training is able to learn policy representations that elevate performance on downstream MARL tasks. We introduce the first offline dataset based on StarCraftII with diverse quality levels and propose a *multi-agent decision transformer* (MADT) for effective offline learning. MADT integrates the powerful temporal representation learning ability of Transformer into both offline and online multi-agent learning, which promotes generalisation across agents and scenarios. The proposed method demonstrates superior performance than the state-of-the-art algorithms in offline MARL. Furthermore, when applied to online tasks, the pre-trained MADT largely improves sample efficiency, even in zero-shot task transfer. To our best knowledge, this is the first work to demonstrate the effectiveness of pre-trained models in terms of sample efficiency and generalisability enhancement in MARL.

## 1 INRTODUCTION

Multi-Agent reinforcement learning (MARL) plays an essential role for solving complex decision-making tasks by learning from the interaction data between machine or autonomous agents and simulated physical environments. It typically applies to self-driving (Shalev-Shwartz et al., 2016), traffic control (Bazzan, 2009), and recommendation system (Xian et al., 2019), and, to specifically mention, surpasses human beings on some video games (Bellemare et al., 2013; Wang et al., 2020). However, the experience-based policy learning scheme requires the algorithms with high sample efficiency because of the limited computing resources and high cost resulting from the data collection (Haarnoja et al., 2018; Munos et al., 2016; Espeholt et al., 2019; 2018). Furthermore, even in domains where the online environment is feasible, we might still prefer to utilize previously collected data instead, for example, if the domain's complex requires large datasets for effective generalization. In addition, a policy trained on one scenario usually cannot perform well on another even though under the same task. Therefore, a universal policy is critical for saving the training time of general reinforcement learning. Offline RL (Levine et al., 2020) attracts more researchers motivated by improving online reinforcement learning with offline datasets (Fu et al., 2020). Those methods like CQL (Kumar et al., 2020), BEAR (Kumar et al., 2019), and BCQ (Fujimoto et al., 2019) conventionally leverage the current off-policy RL algorithms, e.g., DQN (Mnih et al., 2015), with provable constraints (Rashidinejad et al., 2021). They mainly focus on solving the overestimation of Q values learned in offline datasets and the distribution shift between offline and online. However, those constraints limit the performance with insufficiently offline dataset exploitation (Kumar et al., 2019). Recently, the transformer shows its advantage of representing the sequential data for the classification or prediction tasks. Most related to our work, Chen et al. (2021) replace the constrained Q networks in conventional offline RL algorithms with causal transformer and demonstrate the superiority comparing with baselines such as Behavior Cloning (BC) and state-of-the-art offline algorithms in single-agent offline RL. Nevertheless, the overestimation problems emerge in the offline multi-agent RL when one directly applies the sequential modeling method. We propose leveraging

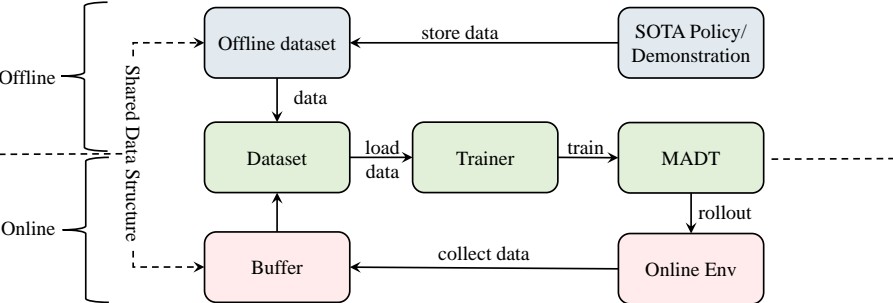

Figure 1: Overview of the pipeline for pre-training the general policy and fine-tuning it online.

the representation capability of the transformer and then load the pre-trained model as part of the policy in the fine-tuning phase.

This paper proposes Multi-Agent Decision Transformers (MADT) for pre-training the general policy on offline datasets, capable of generalizing onto other seen/unseen environments. Firstly, we collect the offline datasets from the well-known, challenging MARL environment, StarCraft Multi-Agent Challenge (SMAC) (Samvelyan et al., 2019), interacting with the state-of-the-art algorithm, multi-agent PPO (Yu et al., 2021). Then we construct transformer variants for multiple agents with parameter sharing or acing as a part. We give the paradigm of pre-training a model with the transformer and fine-tuning it with MARL. We conclude the main contribution as follows:

- We propose a series of transformer variants for offline MARL via leveraging the sequential modeling of the attention mechanism.

- We derive a framework with offline pretraining and online fine-tuning to improve sample efficiency and generality across different scenarios.

- We are the first to apply the pre-trained model across different multi-agent scenarios to validate the transferring ability. Experimental results show fast adaptation and superior performance.

- We build a dataset with different skill-level covering all SMAC scenarios for benchmarking progress in MARL pre-training.

## 2 PRELIMINARY

**Multi-Agent Reinforcement Learning.** For the Markov Game, which is a multi-agent extension from the Markov Decision Process (MDP), there is a tuple representing the essential elements $< \mathbb{S}, \mathbb{A}, R, P, n, \gamma >$, where $\mathbb{S}$ denotes the state space of $n$ agents : $S_1 \times S_2 \ldots S_n \rightarrow \mathbb{S}$. $\mathbb{A}_i$ is the action space of each agent $i$, $P : S_i \times \mathbb{A}_i \rightarrow PD(S_i)$ denotes the transition function emiting the distribution over the state space and $\mathbb{A}$ is the joint action space, $R_i : \mathbb{S} \times \mathbb{A}_i \rightarrow \mathbb{R}$ is the reward function of each agent and takes action following their policies $\pi(a|s) \in \Pi_i : \mathbb{S} \rightarrow PD(\mathbb{A})$ from the policy space $\Pi_i$, where $\Pi_i$ denotes the policy space of agent $i$, $a \sim \mathbb{A}_i$, and $s \sim S_i$. Each agent aims to maximize its long-term reward $\sum_t \gamma^t r_t$, where $r_i^t \in R_i$ denotes the reward of agent $i$ in time $t$ and $\gamma$ denotes the discount factor.

**Attention-based Model.** Attention-based model has shown its stable and strong representation capability. The scale dot-production attention uses the self-attention mechanism demonstrated in (Vaswani et al., 2017). Let $\mathbf{Q} \in \mathbb{R}^{t_q \times d_q}$ be the quries, $\mathbf{K} \in \mathbb{R}^{t_k \times d_k}$ be the keys, and $\mathbf{V} \in \mathbb{R}^{t_v \times d_v}$ be the values, where $t_*$ are the element numbers of different inputs and $d_*$ are the corresponding element dimensions. Normally, $t_k = t_v$ and $d_q = d_k$. The outputs of self-attention are computed as:

$$\text{Attention}(\mathbf{Q}, \mathbf{K}, \mathbf{V}) = \text{softmax}(\frac{\mathbf{Q}\mathbf{K}^T}{\sqrt{d_k}})\mathbf{V} \tag{1}$$

where the scalar $1/\sqrt{d_k}$ is used to prevent the softmax function into regions that has very small gradients. Then we introduce the multi-head attention process as follows:

$$\text{MultiHead}(\mathbf{Q}, \mathbf{K}, \mathbf{V}) = \text{Concat}(\text{head}_1, \ldots, \text{head}_h)\mathbf{W}^O \tag{2}$$

$$\text{head}_i = \text{Attention}(\mathbf{Q}\mathbf{W}_i^Q, \mathbf{K}\mathbf{W}_i^K, \mathbf{V}\mathbf{W}_i^V) \tag{3}$$

The position-wise feed-forward network is another core module of the transformer. It consists of two linear transformations with a ReLU activation in between. The dimensionality of inputs and outputs is $d_{model}$, and that of the feef forward layer is $d_{ff}$. Specially,

$$\text{FFN}(x) = \max(0, x\mathbf{W}_1 + \mathbf{b}_1)\mathbf{W}_2 + \mathbf{b}_2 \tag{4}$$

where $\mathbf{W}_1 \in \mathbb{R}^{d_{model} \times d_{ff}}$ and $\mathbf{W}_2 \in \mathbb{R}^{d_{ff} \times d_{model}}$ are the weights, and $\mathbf{b}_1 \in \mathbb{R}^{d_{ff}}$ and $\mathbf{b}_2 \in \mathbb{R}^{d_{model}}$ are the biases. Across different positions are the same linear transformations. Note that the position encoding for leveraging the order of the sequence as follows:

$$\text{PE}(pos, 2i) = \sin pos/10000^{2i/d_{model}}$$
$$\text{PE}(pos, 2i+1) = \cos pos/10000^{2i/d_{model}} \tag{5}$$

## 3 METHODOLOGY

In this section, we demonstrate how the transformer is applied to our offline pre-training MARL framework. Firstly, we introduce an offline MARL method, in which the transformer maps between the local observations and actions of each agent in the offline dataset via parameter sharing sequentially. Then we leverage the hidden representation as the input of the multi-agent decision transformer (MADT) to minimize the cross-entropy loss. Furthermore, we introduce how to integrate the online MARL with MADT in constructing our whole framework to train a universal MARL policy. To accelerate the online learning, we load the pre-trained model as a part of MARL algorithms and learn the policy based on experience in the latest buffer stored from the online environment. To train a universal MARL policy quickly adapting to other tasks, we bridge the gap between different scenarios from observations, actions, and available actions, respectively. Figure 1 overviews our method from the perspective of offline pre-training with supervised learning and online fine-tuning with MARL algorithms.

### 3.1 MULTI-AGENT DECISION TRANSFORMER

Algorithm 1 shows the training process of Multi-Agent Decision Transformer (MADT), in which we autoregressively encode the trajectories from the offline datasets in offline pre-trained MARL and train the transformer-based network with supervised learning. We carefully reformulate the trajectories as the inputs of the causal transformer different from those in the Decision Transformer (Chen et al., 2021). Denote that we deprecate the reward-to-go and actions that are encoded with states together in the single-agent DT. We will interpret the reason for this in the next section. Similar to the seq2seq models, MADT is based on the autoregressive architecture with the reformulated sequential inputs across timescale. The left part of Figure 2 shows the architecture. The causal transformer encodes the agent $i$'s trajectory sequence $\tau_i^t$ at the time step $t$ to a hidden representation $\mathbf{h}_i^t = (h_1, h_2, \ldots, h_l)$ with a dynamic mask. Given $\mathbf{h}_t$, the output at the time step $t$ only based on the previous data then consumes the previously emitted actions as additional inputs when predicting a new action.

---

**Algorithm 1: MADT-Offline**: Multi-Agent Decision Transformer

---

**Input:** Offline dataset $\mathcal{D} : \{\tau_i : \langle s_i^t, o_i^t, a_i^t, v_i^t, d_i^t, r_i^t \rangle_{t=1}^T\}_{i=1}^n$, $v_i^t$ denotes the available action
**Initialize** $\theta$ for the Causal Transformer;
**Initialize** $\alpha$ as the learning rate, $C$ as the context length, and $n$ as the maximize agent number
**for** $\tau = \{\tau_1, \ldots, \tau_i, \ldots, \tau_n\}$ *in* $\mathcal{D}$ **do**
    Chunk the trajectory into $\tau_i = \{r_i^t, s_i^t, a_i^t\}_{t \in 1:C}$ as the ground truth samples, where $C$ is the context length, and mask the trajectory when $d_i^t$ is true
    **for** $\tau_i^t = \{\tau_i^1 \ldots \tau_i^t \ldots \tau_i^T\}$ *in* $\tau_i$ **do**
        Mask illegal actions via $P(a_j | \tau_i, \hat{a}_{<j}; \theta') = 0$ if $v_i^t$ is True
        Predict the action $\hat{a}_t = \arg\max_a P(a | \tau_i, \hat{a}_{<t}; \theta')$
        Update $\theta$ with $\theta = \arg\max_{\theta'} \frac{1}{C} \sum_{j=1}^{C} P(a_j) \log P(\hat{a}_j | \tau_i, \hat{a}_{<j}; \theta')$
    **end**
**end**
**return** $\theta$

---

**Trajectories Reformulation as Input.** We model the lowest granularity at each time step as a modeling unit $x_t$ from the static offline dataset for the concise representation. MARL has many

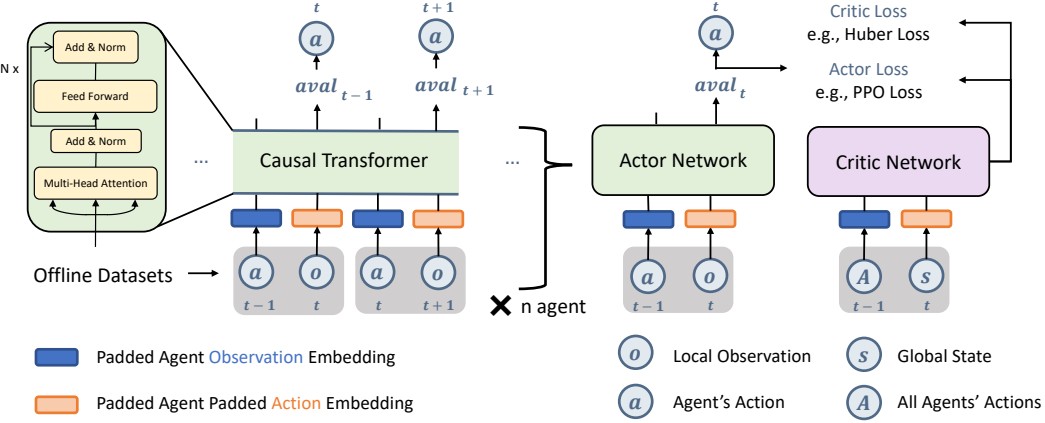

Figure 2: The detailed model structure for offline and online MADT.

elements such as $\langle$global_state, local_observation$\rangle$, different from the single agent. It is reasonable for sequential modeling methods to model them in Markov Decision Process (MDP). Therefore, we formulate the trajectory as follows:

$$\tau^i = (x_1, \ldots x_t \ldots, x_T) \quad \text{where } x_t = (s_t, o_t^i, a_t^i)$$

where $s_t^i$ denotes the global shared state, $o_t^i$ denotes the individual observation for agent $i$ at time step $t$, and $a_t^i$ denotes the action. We take $x_t$ as a token similar to the input of natural language processing.

**Output Sequence Constructing.** To bridge the gap between training with the whole context trajectory and testing with only previous data, we mask the context data to autoregressively output in the time step $t$ with previous data in $\langle 1 \ldots t-1 \rangle$. Therefore, MADT predicts the sequential actions at each time step using the decoder as follows:

$$y = a_t = \arg\max_a p_\theta(a|\tau, a_1, \ldots, a_{t-1}) \tag{6}$$

where $\theta$ denotes the parameter of MADT and $\tau$ denotes the trajectory including the global state $s$, local observation $o$ before the time step $t$, $p_\theta$ is the distribution over the legal action space under the available action $v$.

**Core Module Description.** MADT differs from the transformers in conventional sequence modeling tasks that take inputs with position encoding and decode the encoded hidden representation autoregressively. We use the masking mechanism with a lower triangular matrix to compute the attention:

$$\text{Attention}(\mathbf{Q}, \mathbf{K}, \mathbf{V}) = \text{softmax}(\frac{\mathbf{Q}\mathbf{K}^T}{\sqrt{d_k}} + M)\mathbf{V} \tag{7}$$

where $M$ is the mask matrix which ensures the input at the time step $t$ can only correlate with the input from $\langle 1, \ldots, t-1 \rangle$. We employ the Cross-entropy (CE) as the total sequential prediction loss and utilize the available action $v$ to ensure agents taking those illegal actions with probability zero. The CE loss can be represented as follows:

$$L_{CE}(\theta) = \frac{1}{C} \sum_{t=1}^{C} P(a_t) \log P(\hat{a}_t|\tau_t, \hat{a}_{<t}; \theta), \quad \text{where } C \text{ is the context length} \tag{8}$$

where $a_t$ is the ground truth action, $\tau_t$ includes $\{s_{1:t}, o_{1:t}\}$. $\hat{a}$ denotes the output of MADT. The cross-entropy loss shown above aims to minimize the distribution distance between the prediction and the ground truth.

## 3.2 MULTI-AGENT DECISION TRANSFORMER WITH PPO

The method above can fit the data distribution well resulting from the sequential modeling capacity of the transformer. But it fails to work well when pre-training on the offline datasets and improving continually by interacting with the online environment. The reason originates from the mismatch between the objective in MADT and ignore the value of each action. When the pre-trained model

is loaded to interact with the online environment, the buffer will only collect actions conforming to the distributions of the offline datasets rather than those corresponding to high reward at this state. That means the pre-trained policy is regarded as good once the selected action is identical to that in the dataset, even though it leads to a low reward. Therefore, we need to design another paradigm, MADT-PPO, to integrate RL and supervised learning for fine-tuning continually in Algorithm 2. Figure 2 shows the pre-training and fine-tuning framework. A direct method is to share the pre-trained model across each agent and implement the REINFORCE algorithm (Williams, 1992). However, only actors result in higher variance, and the employment of a critic to assess state values is necessary. Therefore, in online MARL, we leverage an extension of PPO, the state-of-the-art algorithm on tasks of StarCraft, MPE, and even return-based game Hanabi (Mordatch & Abbeel, 2018).

---

**Algorithm 2: MADT-Online**: Multi-Agent Decision Transformer with PPO

---

**Input:** Offline dataloader $\mathcal{D}$, Pretrained MADT Policy with parameter $\theta$
**Initialize** $\theta$ and $\phi$ are the parameters of an actor $\pi_\theta(a_i|o_i)$ and critic $V_\phi(s)$ respectively, which could be inherited directly from pre-trained models.
**Initialize** $n$ as the agent number, $\gamma$ as the discount factor, and $\epsilon$ as clip ratio
**for** $\tau = \{\tau_1, \ldots, \tau_i, \ldots, \tau_n\}$ *in* $\mathcal{D}$ **do**
    Sample $\tau_i = \{s^t, o_i^t, a_i^t\}_{t \in 1:T}$ as the ground truth, where $T$ is the context length
    Compute the advantage function $A(s, a_i) = \sum_t \gamma^t r(s, a_i) - V_\phi(s)$
    Computing the important weight $w = \pi_\theta(o_i, a_i)/\pi_{\theta_{old}}(o_i, a_i)$
    Update $\theta_i$ for $i \in 1 \ldots n$ via:
        $\theta_i = \arg\max \mathbb{E}_{s \sim \rho_{\theta_{old}}, a \sim \pi_{\theta_{old}}}[clip(w, 1 - \epsilon, 1 + \epsilon)A(s, a_i)]$
    Compute the MSE loss $L_\phi = \frac{1}{2}[\sum_t \gamma^t r - V_\phi(s)]^2$
    Update the critic network via $\phi = \arg\min_\phi L_\phi$
**end**
**return** $\theta, \phi$

---

### 3.3 UNIVERSAL MODEL ACROSS SCENARIOS

To train a universal policy for each of the scenarios in the SMAC which might be vary in agent number, feature space, action space and reward ranges, we consider the modification list below.

**Parameters Sharing Across Agents.** When offline examples are collected from multiple tasks or the test phase owns the different agent numbers from the offline datasets, the difference of agent number across tasks is an intractable problem for deciding the number of actors. Thus, we consider sharing the parameters across all actors with one model as well as attaching one-hot agent IDs into observations, for compatibility with a variable number of agents.

**Feature Encoding.** When the policy needs to generalize to new scenarios that arises from different feature shapes, We propose encoding all features into a universal space by padding zero at the end and mapping to a low-dimensional space with fully connected networks.

**Action Masking.** Another issue is the different action space across scenarios. For example, less enemies in a scenario means less potential attack options as well as available actions. Therefore, an extra vector is utilized to mute the unavailable actions so that their probabilities are always zero during both learning and evaluating process.

**Reward Scaling.** Different scenarios might be vary in reward ranges and lead to unbalanced models during multi-task offline learning. To balance the influence of examples from different scenarios, we scale their rewards into the same range to ensure the output models have comparable performance between different tasks.

## 4 EXPERIMENTS

We show three experimental settings: offline MARL, online MARL by loading the pre-trained model, and few-shot or zero-shot offline learning. For the offline MARL, we expect to verify the performance of our method in pre-training the policy and directly testing on the corresponding maps. For the fine-tuning in the online environment, we aim to demonstrate the capacity of the pre-trained policy on the original or new scenarios. Experimental results in offline MARL shows our MADT-offline

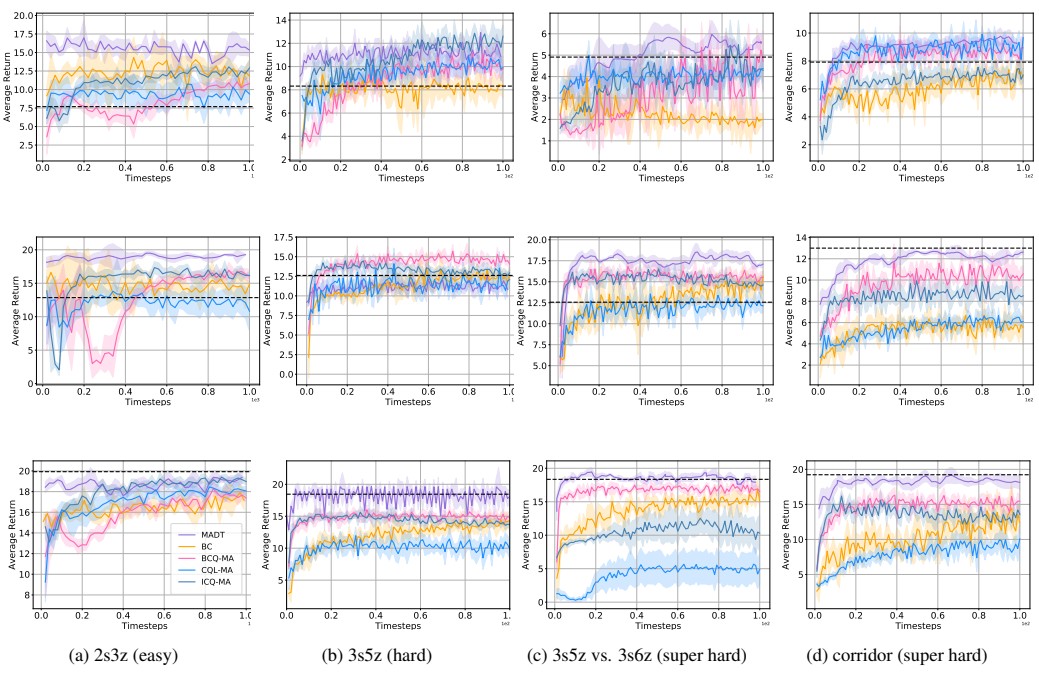

Figure 3: Performance of offline MADT comparing with baselines on four easy or (super-)hard SMAC maps. The dotted lines represent the mean values in the training set. Columns (a-d) are average returns from (poor, medium, good) datasets from top to the bottom.

in 3.1 outperforms the state-of-the-art methods. Furthermore, MADT-online in 3.2 can improve the sample efficiency across multiple scenarios. Besides, the universal MADT trained from multi-task data with MADT-online generalize well in each scenario in few-shot even zero-shot setting.

### 4.1 OFFLINE DATASETS

The offline datasets are collected from the running policy, MAPPO (Yu et al., 2021), on the well-known SMAC task (Samvelyan et al., 2019). Each dataset contains a large number of trajectories: $\tau := (s_t, o_t, a_t, r_t, done_t, v_t)_{t=1}^T$. Different from D4RL (Fu et al., 2020), our datasets consider the property of DecPOMDP, which owns local observations and available actions for each agent. In Appendix, we list the statistical properties of the offline datasets in Table 2 and Table 3.

### 4.2 OFFLINE MULTI-AGENT REINFORCEMENT LEARNING

In this experiment, we aim to validate the effectiveness of MADT offline version in Section 3.1 as a framework for offline MARL on the static offline datasets. We train a policy on the offline datasets with various qualities and then apply it to an online environment, StarCraft (Samvelyan et al., 2019). There are also baselines under this setting, such as Behavior Cloning (BC) as a kind of imitation learning method showing stable performance on single-agent offline RL. Besides, we employ the conventional effective single-agent offline RL algorithms, BCQ (Fujimoto et al., 2019), CQL (Kumar et al., 2020), and ICQ (Yang et al., 2021), then add the mixing network for multi-agent setting, denoting as "xx-MA". To verify the quality of our collected datasets, we choose data from different level and train the baselines as well as our MADT. Figure 3 shows the overall performance on various quality datasets. The baseline methods enhance their performance stably, indicating the quality of our offline datasets. The results also show that our MADT outperforms the offline MARL baselines and converges faster across easy, hard, and super hard maps (2s3z, 3s5z, 3s5z_vs_3s6z, corridor).

### 4.3 OFFLINE PRE-TRAINING AND ONLINE FINE-TUNING

The experiment designed in this subsection intends to answer the question: **Is the pre-training process is necessary for online MARL?** Firstly we compare the online version of MADT in Section 3.2 with and without loading the pre-trained model. If training MADT only by online experience, we can view it as a transformer-based MAPPO replacing the actor and critic backbone networks with the

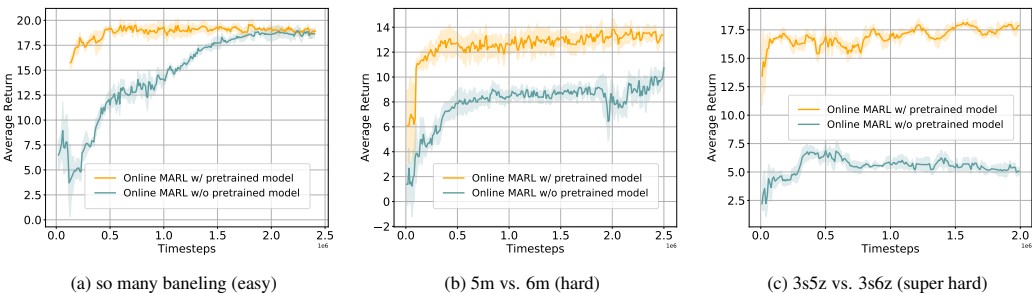

|                          | (a) so many baneling (easy) | (b) 5m vs. 6m (hard) | (c) 3s5z vs. 3s6z (super hard) |

Figure 4: The average returns with and without the pre-trained model..

transformer. Furthermore, we validate that our framework MADT with the pre-trained model can improve sample efficiency on most easy, hard, and super hard maps.

**Necessity of Pretrained Model.** We train our MADT based on the datasets collected from a map and fine-tune it on the same map online with the MAPPO algorithm. For comparison fairness, we use the transformer as both actor and critic networks with and without the pre-trained model. Primarily, we choose three maps from Easy, Hard, and Super Hard maps to validate the effectiveness of the pre-trained model in Figure 4. Experimental results show that the pre-trained model converges faster than the algorithm trained from scratch, especially in the challenging maps.

**Improving Sample Efficiency.** For validating the improvement of the sample efficiency via loading our pre-trained MADT and fine-tuning it with MAPPO, we compare the overall framework with the state-of-the-art algorithm, MAPPO (Yu et al., 2021), without the pre-training phase. We measure the number of interactions with the environment in Table 1, and our pre-trained model needs much less than the traditional MAPPO to access the same win rate.

| Maps | # Samples to Acess the Win Rate | | | | | Maps | # Samples to Acess the Win Rate | | | | |
|---|---|---|---|---|---|---|---|---|---|---|---|
| | 20% | 40% | 60% | 80% | 100% | | 20% | 40% | 60% | 80% | 100% |
| 2m vs. 1z (Easy) | 8e4/- | 1e5/- | 1.3e5/- | 1.5e5/- | 1.6e5/- | 3s vs. 5z (Hard) | 8e5/- | 8.5e5/- | 8.7e5/1.5e4 | 9e5/5e4 | 2e6/1.5e5 |
| 3m (Easy) | 3.2e3/- | 8.3e4/- | 3.2e5/- | 4e5/- | 7.2e5/- | 2c vs. 64 zg (Hard) | 2e5/- | 3e5/- | 4e5/8e4 | 5e5/1e5 | 1.8e6/5e5 |
| 2s vs 1sc (Easy) | 1e4/- | 2.5e4/- | 3e4/- | 8e4/4e4 | 3e5/1.2e5 | 8m vs. 9m (Hard) | 3e5/- | 6e5/- | 1.4e6/2e4 | 2e6/8e4 | ∞/2.2e6 |
| 3s vs. 3z (Easy) | 2.5e5/- | 3e5/- | 6.2e5/1e4 | 7.3e5/1.5e5 | 8e5/2.9e5 | 5m vs. 6m (Hard) | 1.5e6/2e5 | 2.5e6/8e5 | 5e6/2e6 | ∞/∞ | ∞/∞ |
| 3s vs. 4z (Easy) | 3e5/- | 4e5/- | 5e5/- | 6.2e5/- | 1.5e6/1.8e5 | 3s5z (Hard) | 8e5/6.3e4 | 1.3e6/1e5 | 1.5e6/4e5 | 1.9e6/1e6 | 2.5e6/2e6 |
| so many baneling (Easy) | 3.2e4/8e3 | 1e5/4e4 | 3.2e5/7e4 | 5e5/8e4 | 1e6/6.4e5 | 10m vs. 11m (Hard) | 2e5/- | 3.5e5/- | 4e5/2.8e4 | 1.7e6/1.2e5 | 4e6/2.5e5 |
| 8m (Easy) | 4e5/- | 5e6/- | 5.6e5/- | 5.6e5/1.6e5 | 8.8e5/2.4e5 | MMM2 (Super Hard) | 1e6/ | 1.8e6/ | 2.3e6/ | 4e6/ | ∞/∞ |
| MMM (Easy) | 5.2e4/- | 8e4/- | 3e5/- | 4.5e5/- | 1.8e6/6e5 | 3s5z vs. 3s6z (Super Hard) | 1.8e6/- | 2.5e6/- | 3e6/8e5 | 5e6/1e6 | ∞/∞ |
| bane vs. bane (Easy) | 3.2e3/- | 3.2e3/- | 3.2e5/- | 4e5/- | 5.6e5/- | corridor (Super hard) | 1.5e6/- | 1.8e6/- | 2e6/- | 2.8e6/- | 7.8e6/4e5 |

Table 1: The sample numbers needed to access the win rate 20%, 40%, 60%, 80%, and 100% with (MAPPO/pre-trained MADT). "-" represents that no more samples is needed to reach the target win rate. "∞" represents that policies cannot reach the target win rate.

## 4.4 GENERALIZTION WITH MULTI-TASK PRE-TRAINING

Experiments in this section are to explore the transferability of the universal MADT mentioned in Section 3.3, which is pre-trained with mixed data from multiple tasks. According to whether the downstream tasks have been seen or not, the few-shot experiments are designed to validate the adaptibilty on seen tasks, while the zero-shot experiments are designed for the held-out maps.

**Few-shot Learning.** The results in Figure 5a shows that our method can utilize multi-task datasets to train a universal policy and generalize to all tasks well. Pre-trained MADT can achieve higher return than the model trained from scratch when we limit the interactions with environment.

**Zero-shot Learning.** Figure 5b shows that our universal MADT can surprisingly improve performance on downstream task even if it has not been seen before. (3 stalkers vs. 4 zealots).

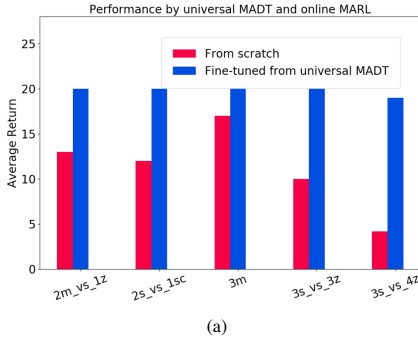 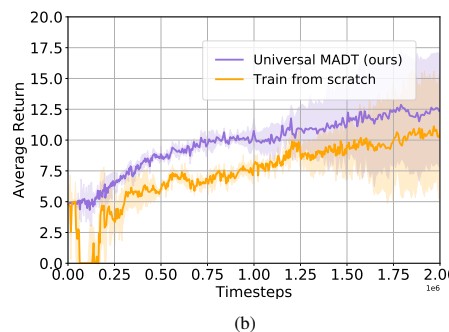

(a) (b)

Figure 5: Few-shot and Zero-shot validation results. (a) shows the average returns of the universal MADT pre-trained from all five tasks data and the policy trained from scratch, individually. We limit the environment interaction to 2.5M steps. (b) shows the average returns of a held-out map (`3s_vs_4z`), where the universal MADT is trained from data on (`2m_vs_1z, 2s_vs_1sc, 3m, 3s_vs_3z`).

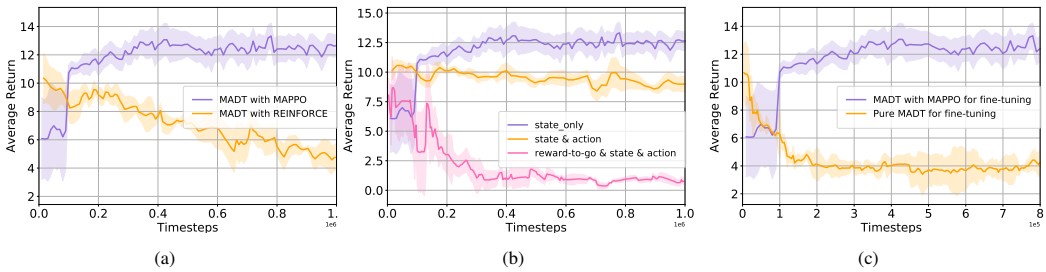

(a) (b) (c)

Figure 6: Ablation results on a hard map, `5m_vs_6m`, for validating the necessity of (a) MAPPO in MADT-Online, (b) input formulation, (c) online version of MADT.

## 4.5 ABLATION STUDY

The experiments in this subsection are designed to answer the following research questions: **RQ1:** Why we choose MAPPO for online phase? **RQ2:** Which kind of input shall we use to make the pre-trained model beneficial for the online MARL? **RQ3:** Why cannot the offline version of MADT be improved in the online fine-tune period after pre-training?

**Suitable Online Algorithm.** Although the selection of MARL algorithm for online phase should be flexible according to specific task, we design experiments to answer **RQ1** here. As discussed in Section 3, we can train Decision Transformer for each agent and online fine-tune it with a MARL algorithm. An intuitive method is to load the pre-trained transformer and take it as the policy network for fine-tuning with the policy gradient method, e.g. REINFORCE (Williams, 1992). However, for the reason of high variance mentioned in Section 3.2, we choose MAPPO as the online algorithm and compare their performance in improving sample efficiency during the online period in Figure 6a.

**Dropping Reward-to-go in MADT.** To answer **RQ2**, we compare different inputs embedded into the transformer, including the combination of state, reward-to-go, and action. We find the reward-to-go harmful to online fine-tuning performance, as shown in Figure 6b. We suppose the distribution of reward-to-go is mismatch between offline data and online samples. That is, rewards of online samples are usually lower than offline data due to stochastic exploration at the beginning of the online phase. It deteriorates the fine-tuning capability of the pre-trained model, and based on Figure 6b, we only choose states as our inputs for pre-training and fine-tuning.

**Integrating Online MARL with MADT.** To answer **RQ3**, we directly apply the offline version of MADT for pre-training and fine-tune it online. However, Figure 6c shows that it cannot be improved during the online phase. We analyze the results from the absence of motivation for chasing higher rewards, and conclude that offline MADT is in fact supervised learning and tends to fit its collected experience even with unsatisfactory reward.

## 5 RELATED WORK

**Offline Deep Reinforcement Learning.** Conventionally, offline RL trains a policy based on the static and previously collected data without any environmental interaction (Levine et al., 2020). This policy is then leveraged to interact with the online environment to obtain promising results. A straightforward method for the current reinforcement learning is to use the off-policy algorithm and regard the offline datasets as a replay buffer to learn a policy with good performance. However, experience existing in offline datasets and interaction with online environment have different distributions which cause the overestimation in the off-policy (value-based) method (Kumar et al., 2019). Substantial works presented in offline RL aim at resolving the distribution shift between the static offline datasets and the online environment interaction (Kumar et al., 2020; Fujimoto et al., 2019; Kumar et al., 2019). Yang et al. (2021); Jiang & Lu (2021) constrain off-policy algorithms in offline MARL. Related to our work for the improvement of sample efficiency, Nair et al. (2020) load the pretrained model to generate a weight for actor-critic online RL algorithms. Recently, the Decision Transformer outperforms many state-of-the-art offline RL algorithms via regarding the training process as a sequential modeling phase and test on the online environment (Chen et al., 2021; Janner et al., 2021). In contrast, we show a transformer-based method in the multi-agent field, attempting to transfer across many scenarios without extra constraints.

**Multi-Agent Reinforcement Learning.** As a natural extension from single-agent RL, MARL attracts much attention to solve more complex problems under Markov Games (Zhang et al., 2021). Related algorithms conventionally enforce multiple agents to interact with the environment online and collect the experience to train the joint policy from scratch (Rashid et al., 2018; Sunehag et al., 2017; Son et al., 2019; Mahajan et al., 2019; Yu et al., 2021). The framework of traditional multi-agent learning employs the centralized training and decentralized execution framrwork due to local observation and increasing action space. We build offline datasets by collecting elements from the DecPOMDP and train the general policy with parameter sharing. We utilize transformer to learn a policy with previously collected data from the golden policy in advance and fine-tune it online.

**Transformer.** Transformer (Vaswani et al., 2017) has achieved a great breakthrough to model relations between the input and output sequence with variable length, for the sequence-to-sequence problems (Sutskever et al., 2014), especially on machine translation (Wang et al., 2019) and speech recognition (Dong et al., 2018). Recent works even reorganize the vision problems as the sequential modeling process and construct the SOTA model with pretraining, named ViT (Han et al., 2020; Dosovitskiy et al., 2020; Liu et al., 2021). Due to the Markovian property of trajectories in offline datasets, we can utilize Transformer as that in language modeling. Therefore, Transformer can bridge the gap between supervised learning in the offline setting and reinforcement learning in the online interaction because of the representation capability. We claim that the components under Markov Games are sequential then utilize the transformer for each agent to fit a transferable MARL policy. Furhtermore, we fine-tune the learned policy under via trial-and-error.

## 6 CONCLUSIONS

In this work, we propose MADT, an offline pre-trained model for MARL, which integrates the transformer to improve sample efficiency and generalizability. We build an offline MARL dataset based on SMAC. Our method outperforms the state-of-the-art methods in offline MARL, including BC, BCQ, CQL, and ICQ. Our whole framework can also drastically improve the sample efficiency of online MARL via loading pre-trained model. Furthermore, we apply our method to train a universal policy over the multi-task datasets and fine-tune it under the few-shot and zero-shot settings. Results demonstrate that this universal policy adapts fast to new tasks and promises performance on different level maps.

To our best knowledge, we are the first to pre-train the multi-agent model with offline datasets and fine-tune it under the few-shot and zero-shot settings. Future research will pre-train a more general policy, which can transfer across more diverse tasks.

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

# A    PROPERTIES OF DATASETS

We list the properties of our offline datasets in Tables 2 and 3.

| Maps | Difficulty | Data Quality | # Samples | Reward Distribution (mean ($\pm$std)) |
|---|---|---|---|---|
| **3m** | **Easy** | 3m-poor | 62528 | 6.29 ($\pm$2.17) |
| | | 3m-medium | - | - |
| | | 3m-good | 1775159 | 19.99 ($\pm$0.18) |
| **8m** | **Easy** | 8m-poor | 157133 | 6.43 ($\pm$2.41) |
| | | 8m-medium | 297439 | 11.95 ($\pm$0.94) |
| | | 8m-good | 2781145 | 20.00 ($\pm$0.16) |
| **2s3z** | **Easy** | 2s3z-poor | 147314 | 7.69 ($\pm$1.77) |
| | | 2s3z-medium | 441474 | 12.85 ($\pm$1.37) |
| | | 2s3z-good | 4177846 | 19.93 ($\pm$0.67) |
| **2s_vs_1sc** | **Easy** | 2s_vs_1sc-poor | 12887 | 6.62 ($\pm$2.74) |
| | | 2s_vs_1sc-medium | 33232 | 11.70 ($\pm$0.73) |
| | | 2s_vs_1sc-good | 1972972 | 20.23 ($\pm$0.02) |
| **3s_vs_4z** | **Easy** | 3s_vs_4z-poor | 216499 | 7.58 ($\pm$1.45) |
| | | 3s_vs_4z-medium | 335580 | 12.13 ($\pm$1.38) |
| | | 3s_vs_4z-good | 3080634 | 20.19 ($\pm$0.40) |
| **MMM** | **Easy** | MMM-poor | 326516 | 7.64 ($\pm$2.05) |
| | | MMM-medium | 648115 | 12.23 ($\pm$1.37) |
| | | MMM-good | 2423605 | 20.08 ($\pm$1.67) |
| **so_many_baneling** | **Easy** | so_many_baneling-poor | 1542 | 9.08 ($\pm$0.66) |
| | | so_many_baneling-medium | 59659 | 13.31 ($\pm$1.14) |
| | | so_many_baneling-good | 1376861 | 19.46 ($\pm$1.29) |
| **3s_vs_3z** | **Easy** | 3s_vs_3z-poor | 52807 | 8.10 ($\pm$1.37) |
| | | 3s_vs_3z-medium | 80948 | 11.87 ($\pm$1.19) |
| | | 3s_vs_3z-good | 149906 | 20.02 ($\pm$0.09) |
| **2m_vs_1z** | **Easy** | 2m_vs_1z-poor | 25333 | 5.20 ($\pm$1.66) |
| | | 2m_vs_1z-medium | 300 | 11.00 ($\pm$0.01) |
| | | 2m_vs_1z-good | 120127 | 20.00 ($\pm$0.01) |
| **bane_vs_bane** | **Easy** | bane_vs_bane-poor | 63 | 1.59 ($\pm$3.56) |
| | | bane_vs_bane-medium | 3507 | 14.00 ($\pm$0.93) |
| | | bane_vs_bane-good | 458795 | 19.97 ($\pm$0.36) |
| **1c3s5z** | **Easy** | 1c3s5z-poor | 52988 | 8.10 ($\pm$1.65) |
| | | 1c3s5z-medium | 180357 | 12.68 ($\pm$1.42) |
| | | 1c3s5z-good | 2400033 | 19.88 ($\pm$0.69) |

Table 2: The properties for our offline dataset collected from the experience of multi-agent PPO on the easy maps of SMAC.

| Maps | Difficulty | Data Quality | # Samples | Reward Distribution (mean ($\pm$std)) |
|---|---|---|---|---|
| **5m_vs_6m** | **Hard** | 5m_vs_6m-poor | 1324213 | 8.53 ($\pm$1.18) |
| | | 5m_vs_6m-medium | 657520 | 11.03 ($\pm$0.58) |
| | | 5m_vs_6m-good | 503746 | 20 ($\pm$0.01) |
| **10m_vs_11m** | **Hard** | 10m_vs_11m-poor | 140522 | 7.64 ($\pm$2.39) |
| | | 10m_vs_11m-medium | 916845 | 12.72 ($\pm$1.25) |
| | | 10m_vs_11m-good | 895609 | 20 ($\pm$0.01) |
| **2c_vs_64zg** | **Hard** | 2c_vs_64zg-poor | 10830 | 8.91 ($\pm$1.01) |
| | | 2c_vs_64zg-medium | 97702 | 13.05 ($\pm$1.37) |
| | | 2c_vs_64zg-good | 2631121 | 19.95 ($\pm$1.24) |
| **8m_vs_9m** | **Hard** | 8m_vs_9m-poor | 184285 | 8.18 ($\pm$2.14) |
| | | 8m_vs_9m-medium | 743198 | 12.19 ($\pm$1.14) |
| | | 8m_vs_9m-good | 911652 | 20 ($\pm$0.01) |
| **3s_vs_5z** | **Hard** | 3s_vs_5z-poor | 423780 | 6.85 ($\pm$2.00) |
| | | 3s_vs_5z-medium | 686570 | 12.12 ($\pm$1.39) |
| | | 3s_vs_5z-good | 2604082 | 20.89 ($\pm$1.38) |
| **3s5z** | **Hard** | 3s5z-poor | 365389 | 8.32 ($\pm$1.44) |
| | | 3s5z-medium | 2047601 | 12.61 ($\pm$1.32) |
| | | 3s5z-good | 1448424 | 18.45 ($\pm$2.03) |
| **3s5z_vs_3s6z** | **Super Hard** | 3s5z_vs_3s6z-poor | 594089 | 7.92 ($\pm$1.77) |
| | | 3s5z_vs_3s6z-medium | 2214201 | 12.56 ($\pm$1.37) |
| | | 3s5z_vs_3s6z-good | 1542571 | 18.35 ($\pm$2.04) |
| **27m_vs_30m** | **Super Hard** | 27m_vs_30m-poor | 102003 | 7.18 ($\pm$2.08) |
| | | 27m_vs_30m-medium | 456971 | 13.19 ($\pm$1.25) |
| | | 27m_vs_30m-good | 412941 | 17.33 ($\pm$1.97) |
| **MMM2** | **Super Hard** | MMM2-poor | 1017332 | 7.87 ($\pm$1.74) |
| | | MMM2-medium | 1117508 | 11.79 ($\pm$1.28) |
| | | MMM2-good | 541873 | 18.64 ($\pm$1.47) |
| **corridor** | **Super Hard** | corridor-poor | 362553 | 4.91 ($\pm$1.71) |
| | | corridor-medium | 439505 | 13.00 ($\pm$1.32) |
| | | corridor-good | 3163243 | 19.88 ($\pm$0.99) |

Table 3: The properties for our offline dataset collected from the experience of multi-agent PPO on the hard and super hard maps of SMAC.

# B DETAILS OF HYPER-PARAMETERS

Details of hyper-parameters used for MADT experiments are listed from Table 4 to 8.

| Hyper-parameter | Value | Hyper-parameter | Value | Hyper-parameter | Value |
|---|---|---|---|---|---|
| offline_train_critic | True | max_timestep | 400 | eval_epochs | 32 |
| n_layer | 2 | n_head | 2 | n_embd | 32 |
| online_buffer_size | 64 | model_type | state_only | mini_batch_size | 128 |

Table 4: Common hyper-parameters for all MADT experiments

| Maps | offline_episode_num | offline_lr |
|---|---|---|
| 2s3z | 1000 | 1e-4 |
| 3s5z | 1000 | 1e-4 |
| 3s5z_vs_3s6z | 1000 | 5e-4 |
| corridor | 1000 | 5e-4 |

Table 5: Hyper-parameters for MADT experiments in Figure 3

| Maps | offline_episode_num | offline_lr | online_lr | online_ppo_epochs |
|---|---|---|---|---|
| 2c_vs_64zg | 1000 | 5e-4 | 5e-4 | 10 |
| 10m_vs_11m | 1000 | 5e-4 | 5e-4 | 10 |
| 8m_vs_9m | 1000 | 1e-4 | 5e-4 | 10 |
| 3s_vs_5z | 1000 | 1e-4 | 5e-4 | 10 |
| 3s5z | 1000 | 1e-4 | 5e-4 | 10 |
| 3m | 1000 | 1e-4 | 5e-4 | 15 |
| 2s_vs_1sc | 1000 | 1e-4 | 5e-4 | 15 |
| MMM | 1000 | 1e-4 | 1e-4 | 5 |
| so_many_baneling | 1000 | 1e-4 | 1e-4 | 5 |
| 8m | 1000 | 1e-4 | 1e-4 | 5 |
| 3s_vs_3z | 1000 | 1e-4 | 1e-4 | 5 |
| 3s_vs_4z | 1000 | 1e-4 | 1e-4 | 5 |
| bane_vs_bane | 1000 | 1e-4 | 1e-4 | 5 |
| 2m_vs_1z | 1000 | 1e-4 | 1e-4 | 5 |
| 2c_vs_64zg | 1000 | 1e-4 | 1e-4 | 5 |
| 5m_vs_6m | 1000 | 1e-4 | 1e-4 | 10 |
| corridor | 1000 | 1e-4 | 1e-4 | 10 |
| 3s5z_vs_3s6z | 1000 | 1e-4 | 1e-4 | 10 |

Table 6: Hyper-parameters for MADT experiments in Figure 4, 6 and Table 1

| Hyper-parameter | value |
|---|---|
| offline_map_lists | [3s_vs_4z, 2m_vs_1z, 3m, 2s_vs_1sc, 3s_vs_3z] |
| offline_episode_num | [200, 200, 200, 200, 200] |
| offline_lr | 5e-4 |
| online_lr | 1e-4 |
| online_ppo_epochs | 5 |

Table 7: Hyper-parameters for MADT experiments in Figure 5a

| Hyper-parameter | value |
|---|---|
| offline_map_lists | [2m_vs_1z, 3m, 2s_vs_1sc, 3s_vs_3z] |
| offline_episode_num | [250, 250, 250, 250] |
| offline_lr | 5e-4 |
| online_lr | 1e-4 |
| online_ppo_epochs | 5 |

Table 8: Hyper-parameters for MADT experiments in Figure 5b

