# OpenReview forum: "Offline Pre-trained Multi-Agent Decision Transformer"
_ICLR.cc/2022/Conference — ICLR 2022 Submitted_

### Official Review · Reviewer_nCt4 · 2021-10-17

**Correctness:** 3
**Technical Novelty And Significance:** 1
**Empirical Novelty And Significance:** 3
**Recommendation:** 3
**Confidence:** 5

**Main Review:**

The paper is well constructed and the main idea is clear. The experimental evaluation is adequate. I really appreciate the efforts in building the offline SMAC dataset.

The contribution and novelty of this paper, however, are limited. Pre-training the RL model from demonstration is a common technique especially in complex tasks. It is trivial to transfer this technique into the multi-agent field, and MADT does not improve this paradigm for the characteristics in multi-agent environments. The authors stress several times that MADT is the first to pre-train the multi-agent model with offline datasets. I suggest toning down the claim.

Transformer has strong representation ability, however, do you think the action distribution is so complex that MLP model cannot deal with it? In BCQ and CQL, MLP model is able to fit the behavior action distribution.

I notices that PPO is trained from the merged offline and online dataset during the online interaction from Figure 1. If so, the training of PPO is incorrect. PPO is an on-policy algorithm, which requires that the distribution of the experiences for training could be far away from the distribution under the policy. However, there is a large distribution shift between the offline dataset and the online dataset.

For experiments, since MADT fits the action distribution by supervised learning, what is the difference between BC and MADT? I feel confused about the meaning of the X-axis (Timesteps) in Figure 3. I understand that xx-MA means mixing the individual Q values to the joint Q value by a QMIX-style network. However, this part should be elaborated.

I suggest removing section 3.3 to the Appendix since it describes the details about SMAC rather than the proposed method.


**Summary Of The Paper:**

The authors propose MADT which adopts the attention-based model to pre-train the agent policies from the offline dataset and fine-tune the policies by online interaction. The authors also provide an offline SMAC dataset with different skill-level.

**Summary Of The Review:**

The contribution is limited and the main idea has been observed in existing work. The current version cannot reach the borderline of ICLR.

---

### Official Review · Reviewer_mKDU · 2021-11-02

**Correctness:** 3
**Technical Novelty And Significance:** 2
**Empirical Novelty And Significance:** 2
**Recommendation:** 3
**Confidence:** 4

**Main Review:**

Novelty:
This paper is not novel as it seems to be a straightforward combination of transformer and MAPPO.

Presentation:
The charity of the paper needs to be improved especially in experiments. As the main point of the paper is the pre-trained transformer, it is not necessary to spend much space to discuss the experimental results of MADT in the offline RL setting. It would better to discuss the challenge of the setting of offline pre-training with online fine-tuning.

Significance:
It is obvious that using transformer improves the ability of representation, and the results are not surprising. Research papers such as BREMEN and MUSBO from the deployment constrained RL setting are not discussed.

**Summary Of The Paper:**

This paper proposes a multi-agent decision transformer to improve the learning performance in the setting of offline pre-training with online fine-tuning. Experimental results show that it outperforms offline RL algorithms, BCQ, CQL and ICQ in multiple Starcraft II games.

**Summary Of The Review:**

This paper studies a new problem in MARL. However, its main algorithm is not novel and the presentation is not clear. Experimental evaluations are not sufficient, the comparison of MADT and BREMEN is needed. As BREMEN also studies the problem of online tuning RL with offline training. Thus I recommend the rejection.

---

### Official Review · Reviewer_yY9Q · 2021-11-02

**Correctness:** 3
**Technical Novelty And Significance:** 2
**Empirical Novelty And Significance:** 3
**Recommendation:** 5
**Confidence:** 3

**Main Review:**

Use Case: To me the biggest strength of this paper is the use case, it would be very practical for many multi-agent settings to consider efficient deployment in a new environment after pre-training on offline data. I also think that the data set provided is a significant contribution to the community to foster more research in this area. That said, the value of this data is somewhat diminished by being from a trained neural network model rather than humans. It possibly could be interesting to at least add behaviors from different models and different architectures to provide more variety to the behavior policy of each agent. Indeed, optimization in the context of changing behavior policies for each agent is one of the main challenges of multi-agent RL due to the non-stationarity it introduces for each agent, which this paper does not really address or discuss.

MADT Architecture: it is a good idea to apply transformers as an emerging architecture choice for RL problems and it does seem to improve generalization capabilities for the agents. However, it is unclear how these innovations tie to the prior literature and concepts in MARL and offline RL in particular. For example see these surveys "Offline Reinforcement Learning: Tutorial, Review,and Perspectives on Open Problems" Levine et al., 2020 and "A Survey and Critique of Multiagent Deep Reinforcement Learning" Hernandez-Leal et al., 2019. In particular I would have expected to see more discussion of topics like off-policy correction, importance sampling, and over-estimation bias. The imitation learning style objective implemented in this work is not justified in light of this theory of offline RL and clearly is very restrictive in terms of needing a consistently high quality behavior policy to get beneficial results during pre-training. Moreover, these off-policy correction and distributional miss-match problems become even more severe in multi-agent environments as each agent's policy is critical to the environment evolution and as such policies cannot be simply considered in isolation. The proposed approach was not clearly related to the multi-agent literature at all and popular approaches were not compared to even on a theoretical/intuitive level.

Experiments: Another selling point of the paper is that the proposed method does well in comparison to popular approaches for single agent offline RL. However, there is very little theoretical understanding of where these performance gains come from. It seems conceivable that the transformer model alone is what is improving performance and that the way offline training and fine-tuning is done add little to performance. Indeed, transformer models have recently been demonstrated to have advantages for RL in general, so it is unclear if there is any value in the context of this multi-agent use case in particular. It would be very nice to see other proposals for offline training using this transformer architecture as well and validate comparisons to approaches from the multi-agent literature as opposed to only considering single agent approaches applied to multi-agent problems.

Minor: there were a number of typos and grammatical issues throughout the paper.


**Summary Of The Paper:**

This is the first paper to pre-train a multi-agent model with offline datasets and online fine-tune it to a particular task. The authors propose MADT an offline pre-training architecture for multi-agent RL integrating transformers with imitation learning on an a collected data set of offline data. Towards this end, the authors contribute a data set of offline multi-agent interactions collected from running multi-agent PPO on 5 tasks from SMAC. On this data set, the proposed MADT method is shown to outperform popular single agent offline RL algorithms from the literature and excel at fine-tuning a pre-trained offline model.


**Summary Of The Review:**

I really liked a lot about this paper, including the cool and quite practical use case that they consider for the first time. Another strong point is that they provide a data set to jump start research in this area, although its overall value to the community is probably not that high as it is from a trained model and not humans. It is also interesting that the authors show how transformer models can provide value in this setting. The trouble is just that it is unclear how the contributions of this paper are related to the literature on MARL and offline RL, making it necessary to provide more analysis to understand whether these gains are connected to the particular setting considered here or rather hold in general. It is great that this approach can improve on popular algorithms for offline single agent RL, but no algorithms from the multi-agent literature were compared to and there is very little theoretical understanding about the value of particular design choices within the algorithm.  As a result, I lean towards rejection at this time believing that this paper can make a much larger impact to the community following another significant round of revisions.

---

### Official Review · Reviewer_3RgR · 2021-11-04

**Correctness:** 1
**Technical Novelty And Significance:** 2
**Empirical Novelty And Significance:** Not applicable
**Recommendation:** 3
**Confidence:** 4

**Main Review:**

Pros:

- The offline trainig and online tuning setting for multi-agent tasks is new and very interesting.
- Authors show the promise of using a state-of-the-art architecture from the CV or NLP field to RL.
- The paper is also well-written and easy to follow, with a clear introduction of the backgrounds, methods, and experiments.
- Authors conduct extensive experiments in SMAC, and compare against recent state-of-the-art methods including BCQ, CQL, and ICQ. MADT outperforms baselines significantly.

Cons:
- I'm concerned about the appropriateness of MADT-Online based on MAPPO. Although it is in the online fine-tuning stage, from Algorithm 2, MAPPO still learns from the offline dataset D. However, since MAPPO is an on-policy method, I'm concerned about whether it is correct for directly applying an on-policy method without any correction technique?
- Compared to baselines including BCQ, CQL, and ICQ, using the transformer architecture can incur more computation costs.


**Summary Of The Paper:**

The paper studies the offline training and online fine-tuning setting for MARL, and show that it is promising to learn policy representatrions to improve the performance on downstream MARL tasks. The paper proposes to integrate the transformer architecture to improve efficiency and generalization ability. Authors conduct extensive experiments to validate its effectiveness.

**Summary Of The Review:**

The setting is very appealing, and the paper is clearly written, but I'm concerned about the MADT-Online method (see above).

---

### Decision · Program_Chairs · 2022-01-20

**Decision:**

Reject

**Comment:**

The reviewers have raised relevant concerns that preclude acceptance and the authors have not provided a response. At this time, all reviewers concur that this paper should be rejected and I agree.